# Evaluation of Response to Atezolizumab Plus Bevacizumab in Patients with Advanced Hepatocellular Carcinoma Using the Combination of Response Evaluation Criteria in Solid Tumors and Alpha-Fetoprotein

**DOI:** 10.3390/cancers15082304

**Published:** 2023-04-14

**Authors:** Takahiro Kinami, Kei Amioka, Tomokazu Kawaoka, Shinsuke Uchikawa, Shintaro Yamasaki, Masanari Kosaka, Yusuke Johira, Shigeki Yano, Kensuke Naruto, Yuwa Ando, Kenji Yamaoka, Yasutoshi Fujii, Hatsue Fujino, Takashi Nakahara, Atsushi Ono, Eisuke Murakami, Wataru Okamoto, Masami Yamauchi, Daiki Miki, Masataka Tsuge, Michio Imamura, Hiroshi Aikata, Shiro Oka

**Affiliations:** 1Department of Gastroenterology, Graduate School of Biomedical and Health Sciences, Hiroshima University, Hiroshima 734-8551, Japan; kinami@hiroshima-u.ac.jp (T.K.); amioka@hiroshima-u.ac.jp (K.A.); shinuchi@hiroshima-u.ac.jp (S.U.); syamasaki@hiroshima-u.ac.jp (S.Y.); m0607k@hiroshima-u.ac.jp (M.K.); jyusuke9@hiroshima-u.ac.jp (Y.J.); yano0319@hiroshima-u.ac.jp (S.Y.); uzumaki1@hiroshima-u.ac.jp (K.N.); yuwando@hiroshima-u.ac.jp (Y.A.); fujiiyasu@hiroshima-u.ac.jp (Y.F.); fujino920@hiroshima-u.ac.jp (H.F.); nakahara@hiroshima-u.ac.jp (T.N.); atsushi-o@hiroshima-u.ac.jp (A.O.); myamauchi@hiroshima-u.ac.jp (M.Y.); daikimiki@hiroshima-u.ac.jp (D.M.); tsuge@hiroshima-u.ac.jp (M.T.); mimamura@hiroshima-u.ac.jp (M.I.); oka4683@hiroshima-u.ac.jp (S.O.); 2Department of Clinical Oncology, Hiroshima University Hospital, Hiroshima 734-8551, Japan; wokamoto@hiroshima-u.ac.jp; 3Department of Gastroenterology, Hiroshima Prefectural Hospital, Hiroshima 734-8530, Japan; aikatahiroshi@icloud.com

**Keywords:** hepatocellular carcinoma (HCC), atezolizumab plus bevacizumab combination therapy (Atezo + Beva), alpha-fetoprotein (AFP), Response Evaluation Criteria in Solid Tumors (RECIST), overall survival (OS), progression-free survival (PFS)

## Abstract

**Simple Summary:**

Hepatocellular carcinoma (HCC) is reportedly the fifth most common malignancy in the world. Atezolizumab plus bevacizumab combination therapy (Atezo + Beva) was approved in 2020 as the first immune-combined therapy and the standard of care for first-line systemic treatment of unresectable HCC. Alpha-fetoprotein (AFP) is the most widely used serum biomarker in HCC. Although some studies have shown the benefit and safety of Atezo + Beva, there are no reports on when to switch from Atezo + Beva to the next treatment, and this is not yet clear. This study investigated the relationship between radiological response and prognosis in patients assessed as having stable disease (SD) in whom it is necessary to decide whether to continue Atezo + Beva. The results showed that a decreased AFP level may reflect early efficacy, and AFP trends could help determine whether to continue Atezo + Beva in patients assessed as having SD.

**Abstract:**

Atezolizumab plus bevacizumab combination therapy (Atezo + Beva) is currently positioned as the first-line therapy for unresectable hepatocellular carcinoma (u-HCC). It may be difficult to decide whether to continue this treatment if radiological response is assessed as stable disease (SD). Therefore, the relationship between radiological response and prognosis was analyzed. A total of 109 patients with u-HCC and Child–Pugh Score of 5–7 received this treatment. Radiological response was assessed using Response Evaluation Criteria in Solid Tumors (RECIST) and modified RECIST at the first and second evaluations. Of SD patients (n = 71) at the first RECIST evaluation, partial response, SD, and progressive disease (PD) were seen in 10, 55, and 6 patients, respectively, at the second evaluation. On multivariate analysis, in patients with SD at the first RECIST evaluation, a 25% or greater increase in the alpha-fetoprotein (AFP) value from initiation of treatment (odds ratio, 7.38; *p* = 0.037) was the independent factor for PD at the second evaluation. In patients with SD (n = 59) at the second RECIST evaluation, decreased AFP from initiation of treatment (hazard ratio, 0.46; *p* = 0.022) was the independent factor related to progression-free survival on multivariate analysis. AFP trends could help decide the Atezo + Beva treatment strategy.

## 1. Introduction

Hepatocellular carcinoma (HCC) is a major malignancy that is reported to be the fifth most common in the world [1]. HCC commonly occurs in patients with chronic hepatitis or cirrhosis secondary to hepatitis B virus (HBV), hepatitis C virus (HCV), excessive alcohol consumption, or diabetes mellitus [2]. The prognosis of patients with unresectable HCC (u-HCC) is grim [3,4]. As for systemic therapy of u-HCC, sorafenib was approved as the first molecular targeted agent (MTA) in 2009 [5], and lenvatinib was approved as a first-line MTA in Japan in 2018 [6]. In addition, regorafenib, ramucirumab, and ramucirumab were approved as second-line MTAs [7,8,9]. Then, atezolizumab plus bevacizumab combination therapy (Atezo + Beva) was approved in 2020 as the first immune combination therapy [10].

Bevacizumab targets vascular endothelial growth factor (VEGF), which is involved in angiogenesis and tumor growth [11,12]. The IMbrave150 trial confirmed that Atezo + Beva preserved patients’ quality of life and prolonged their survival better than sorafenib [13]. Therefore, Atezo + Beva is now the preferred first-line standard of care for u-HCC [14,15]. We can use these six drugs in multidrug sequential therapy, and drugs should be continued or switched based on the patient’s condition and treatment efficacy. Median progression-free survival (PFS) for patients treated with Atezo + Beva is limited to 6.9 months [16], so it is necessary to switch drugs appropriately when the disease worsens with Atezo + Beva.

However, whether to continue this therapy if the response is assessed as stable disease (SD) is sometimes a clinical question. To improve overall survival (OS), the clinical issue is how to decide whether to continue Atezo + Beva or switch to another therapy based on the radiological response, hepatic reserve, tolerability, and condition of the patient.

Therefore, the predictors of early tumor response to Atezo + Beva were analyzed, and the relationship between radiation response and prognosis was examined in patients treated with Atezo + Beva.

## 2. Materials and Methods

### 2.1. Patients

A total of 158 patients who provided consent to participate in this study underwent treatment with Atezo + Beva for u-HCC at Hiroshima University Hospital from October 2020 to November 2022.

The inclusion criteria for this study were as follows: Child–Pugh score of 5–7; Eastern Cooperative Oncology Group performance status (ECOG PS) of 0 or 1; and underwent at least two radiological response evaluations. A history of prior systemic therapy was not a concern.

Eighteen patients were excluded because of a Child–Pugh score of 8–10, and six patients were excluded because of ECOG PS of 2 or 3. Forty patients did not undergo at least two radiological response evaluations. Due to the above exclusion criteria, a total of 49 patients were excluded. In total, 109 patients met the criteria.

Hepatic reserve was evaluated by the Child–Pugh score and the modified albumin–bilirubin (mALBI) grade.

Patients with positive hepatitis B virus infection (HBV) surface antigen were considered to have HBV-caused HCC, and those with positive anti-hepatitis C virus infection (HCV) antibody were considered to have HCV-caused HCC. Those with both negative HBV surface antigen and negative HCV antibody were considered to have non-viral hepatitis-caused HCC.

### 2.2. Treatments

Patients in the Atezo + Beva group received atezolizumab 1200 mg intravenously plus bevacizumab 15 mg/kg once every 3 weeks. The Common Terminology Criteria for Adverse Events version 5.0 were used to evaluate adverse events (AEs).

The dose of bevacizumab was reduced in cases of drug-related AEs, for example proteinuria and high blood pressure, as necessary based on current dosing guidelines. Atezo + Beva were skipped in cases of drug-related AEs, for example fever, liver damage, fatigue, itching, hepatic encephalopathy, increased ascites, and gastrointestinal perforation. Atezo + Beva were discontinued in cases of unacceptable AEs, such as serious immune-related AEs, interstitial pneumonia, and gastrointestinal bleeding.

Patients continued therapy until death or until one of the following criteria for cessation of therapy was met: worsening liver function, AEs that required termination of treatment, deterioration of Eastern Cooperative Oncology Group performance status to 4, or withdrawal of consent.

### 2.3. Assessment of Treatment Response

As a rule, radiological response assessment by dynamic CT/MRI was performed every 1.5 months, the first radiological response evaluation was performed after 1.5 months, and the second radiological response evaluation was performed 3 months after Atezo + Beva initiation. Radiological response was assessed using Response Evaluation Criteria in Solid Tumors (RECIST) version 1.1 and modified Response Evaluation Criteria in Solid Tumors (mRECIST), and the overall response rate (ORR) and disease control rate (DCR) were calculated based on the radiological response.

If patients obtained a complete response (CR) or a partial response (PR) on RECIST or mRECIST evaluation, they were defined as an objective response (OR). OS was defined as the time from initiation of Atezo + Beva to the time of any cause of death or the last visit. PFS was defined as the time from initiation of Atezo + Beva to the time of radiological progression on RECIST evaluation or any cause of death.

### 2.4. Statistical Analysis

The χ^2^ test or Fisher’s exact test, the Mann–Whitney U test, binomial logistic regression analysis, the Kaplan–Meier method and the log-rank test, and Cox proportional hazards analysis were used for statistical analysis. A *p*-value less of than 0.05 was considered significant. All statistical analyses were carried out using Predictive Analytics Software R version 4.1.2.

## 3. Results

### 3.1. Clinical Characteristics of Participating Patients

The patients’ background characteristics are shown in Table 1. Their median age was 71 years, with 17 cases of HBV, 35 cases of HCV, and 56 cases of NBNC as the cause of HCC. The Child–Pugh score at the initiation of Atezo + Beva was 5 in 61 cases, 6 in 37 cases, and 7 in 11 cases, and the mALBI grade was 1 in 40 cases, 2a in 32 cases, and 2b in 37 cases. First-line and later than second-line were 70 and 39 patients, respectively. Overall, 21 patients had vascular invasion, 38 patients had extrahepatic metastasis, and 5 patients had a relative tumor volume of 50% or more. The BCLC stage was A in 6 cases, B in 52 cases, and C in 51 cases. The median observation period was 12.1 months.

### 3.2. Treatment Response and Survival

The median OS and PFS of the 109 patients included in the study were 21.0 months and 9.5 months, respectively (Figure 1). Table 2 shows the results of radiological response evaluations at the first (1.4 (0.63–3.7) months), second (3.0 (1.6–5.8) months), and best times (1.57 (0.63–11.2) months) by RECIST.

At the first radiological response evaluation, 16 patients (14.7%) had PR, 71 patients (65.1%) had SD, and 22 patients (20.2%) had PD by RECIST evaluation (ORR 14.7%, DCR 79.8%). In addition, 1 patient (0.9%) had CR, 37 patients (33.9%) had PR, 51 patients (46.8%) had SD, and 18 patients (16.5%) had progressive disease (PD) by mRECIST evaluation (ORR 34.8%, DCR 81.6%). Furthermore, at the second radiological response evaluation, the ORR and DCR were 19.3% and 73.4% for RECIST and 36.7% and 68.8% for mRECIST. At the best radiological response evaluation, the ORR and DCR were 35.8% and 84.4%, respectively, for RECIST and 51.4% and 85.3%, respectively, for mRECIST.

### 3.3. OS and PFS for Each Initial Radiological Response

To examine whether radiological response is associated with prognosis, as a responder analysis, OS and PFS were each compared between the two groups (OR and non-OR) at the first and second radiological response evaluations and the best response evaluation by RECIST (Figure 2). The median OS rates by RECIST at the first, second, and best response evaluations were all not reached in the OR group, and 21.0, 20.6, and 20.6 months in the SD group, and 16.8, 16.8, and 11.5 months in the PD group, respectively. At the first radiological response evaluation, there was no significant OS stratification by response (*p* = 0.15). At the second radiological response evaluation and the best response evaluation, there was significant stratification of OS for each response (second radiological response *p* = 0.003, best response evaluation *p* < 0.001). The median PFS rates for RECIST by the first, second, and best response evaluations were 10.7, 14.7, and 16.6 months in the OR group, and 10.5, 10.5, and 8.2 months in the SD group, respectively. At the first and second radiological response evaluations, there was no significant PFS stratification by response (*p* = 0.70, 0.052). At the best response evaluation, there was significant stratification of PFS by response (*p* < 0.001). Therefore, in patients who do not achieve OR with continued Atezo + Beva, treatment modification should be considered.

### 3.4. Rates of and Factors Affecting PD at the Second Radiological Evaluation in Patients with SD at the First Radiological Evaluation by RECIST

Figure 2 shows that obtaining OR at the second radiological response evaluation contributes to longer OS. Next, patients with SD at the first radiological response evaluation were examined. Of SD patients (n = 71) at the first radiological response evaluation, OR, SD, and PD were 10, 55, and 6 patients, respectively, at the second radiological response evaluation.

Table 3 shows analyses of factors related to PD at the second radiological response evaluation in the 71 patients with SD at the first radiological response evaluation. Receiver operating characteristic curve was created using AFP increase rate to determine if it would be PD at the second radiological response evaluation, and a 25% or greater increase was calculated. On univariate analysis, factors contributing to PD included drug discontinuation before the first radiological response evaluation and a 25% or greater increase in AFP values from Atezo + Beva initiation. On multivariate analysis, a 25% or greater increase in AFP values at the first radiological response evaluation from Atezo + Beva initiation (odds ratio, 0.14; 95% confidence interval, 0.021–0.88; *p* = 0.037) was the independent factor related to PD at the second radiological response evaluation on multivariate analysis.

Table 4 shows comparison of the rate of PD at the second radiological response evaluation in view of AFP exceeded or below the reference value (10 ng/mL) at the initiation of Atezo + Beva between the group with a 25% or greater increase in AFP values at the first radiological response evaluation from Atezo + Beva initiation and the other group. The number of patients with a 25% or greater increase in AFP values at the first radiological response evaluation from Atezo + Beva initiation was 9; of them, 4 had PD at the second radiological response evaluation (44.4%). The number of patients without a 25% or greater increase in AFP values at the first radiological response evaluation from Atezo + Beva initiation was 62; of them, 2 had PD at the second radiological response evaluation (3.2%). There was significant stratification of the rate of PD at the second radiological response evaluation (*p* = 0.0017). In the group with AFP that exceeded the reference value (10 ng/ml) at the initiation of Atezo + Beva, the number of patients with a 25% or greater increase in AFP values at the first radiological response evaluation from Atezo + Beva initiation was 8; of them, 3 had PD at the second radiological response evaluation (37.5%). The number of patients without a 25% or greater increase in AFP values at the first radiological response evaluation from Atezo + Beva initiation was 29; of them, 1 had PD at the second radiological response evaluation (3.4%). There was also significant stratification of the rate of PD at the second radiological response evaluation (*p* = 0.026). In the group with AFP below the reference value at the initiation of Atezo + Beva, the number of patients with a 25% or greater increase in AFP values at the first radiological response evaluation from Atezo + Beva initiation was 1, and he had PD at the second radiological response evaluation (100%). The number of patients without a 25% or greater increase in AFP values at the first radiological response evaluation from Atezo + Beva initiation was 33; of them, 1 had PD at the second radiological response evaluation (3.4%). There was close to significance of the rate of PD at the second radiological response evaluation (*p* = 0.059).

The median OS evaluation in 71 patients with SD at the first radiological response evaluation by RECIST was 11.1 months in the group with a 25% or greater increase in AFP values at the first radiological response evaluation from Atezo + Beva initiation and 21.0 months in the other group, respectively (Figure 3), and there was significant stratification of OS by an increased AFP value (*p* = 0.024).

### 3.5. Factors Affecting OS and PFS in SD Patients (n = 59) at the Second Radiological Response Evaluation by RECIST

Figure 2 shows that patients who did not achieve OR with continued Atezo + Beva treatment should be considered for treatment modification. To consider whether Atezo + Beva treatment should be continued if the second radiological response evaluation by RECIST is SD, factors involved in OS and PFS were examined.

In 59 patients with SD at the second radiological response evaluation, there was no independent prognostic factor for OS from Atezo + Beva initiation on univariate analysis (Table 5).

Further, on univariate analysis, factors contributing to PFS included a decrease in AFP values from Atezo + Beva initiation and relative tumor volume at the second radiological response evaluation. A decrease in AFP values from Atezo + Beva initiation (HR, 0.46; 95% confidence interval, 0.23–0.89; *p* = 0.022) was the independent prognostic factor for PFS from Atezo + Beva initiation on multivariate analysis (Table 6).

## 4. Discussion

Currently, Atezo + Beva is positioned as the first-line immunotherapy for u-HCC.

Some recent studies reported the efficacy and safety of Atezo + Beva in Japanese patients with u-HCC in real-world clinical practice [17,18,19].

In HCC, both AFP and DCP are important tumor markers. AFP has been used for diagnosis of HCC and as a marker for therapeutic efficacy, and it was reported that increasing DCP was significantly associated with a poor OS for patients treated with Atezo + Beva [20].

We have also previously reported that AFP is a useful tumor marker for the initial efficacy evaluation of sorafenib, lenvatinib, and Atezo + Beva for HCC [21,22,23].

A previous study also reported that early AFP response is useful to predict response to checkpoint inhibitor therapy with nivolumab and pembrolizumab [24].

Furthermore, it has been reported that AFP cutoffs of a ≥75% decrease or a ≤10% increase measured 6 weeks after starting Atezo + Beva treatment were associated with longer OS and PFS [25].

RECIST is the most common method for assessing the efficacy of other cancers [26]. Several cases of tumor enlargement were observed by RECIST despite the reduced contrast effect of the tumor during Atezo + Beva treatment in this study.

Therefore, it is difficult to determine the efficacy of immunotherapy, and there has been a report of a case in which even PD at the first radiological response evaluation was PR at the best response evaluation by RECIST [27].

The phenomenon of tumor shrinkage after assessment as PD by RECIST during immunotherapy is called pseudoprogression. The mechanism of pseudoprogression is thought to involve activated T cells infiltrating the tumor, causing it to appear enlarged on imaging [28]. In malignant melanoma, where the frequency of pseudoprogression is reported to be around 10%, patients with pseudoprogression had a significantly better survival rate than those with true progression [29]. After the initiation of programmed cell death-1 (PD-1) inhibitor therapy, the patients with decreased levels of serum-specific tumor markers (SSTMs) had significantly better OS and PFS than those with increased levels of SSTMs, and AFP was used in five cases of liver cancer [30].

Therefore, when tumor lesions increase during immunotherapy, it is important to consider the possibility of pseudoprogression and true progression and whether to continue or change treatment based on tumor markers.

In the present study, one patient with PD at the first radiological response evaluation by RECIST achieved CR at the best response evaluation with a pronounced decrease in AFP values and had good OS with 22 cycles of atezolizumab and 9 cycles of bevacizumab.

In addition, many cases were judged to be SD, and it is unclear whether it is better to continue Atezo + Beva or switch to other drug therapy for these cases. Therefore, in the present study, the relationship between radiological response and prognosis in patients who received Atezo + Beva therapy was analyzed.

As mentioned above, there have been several cases of reduced tumor contrast despite an increase in tumor diameter, and since immunotherapy is commonly evaluated by RECIST, attention was paid to RECIST. In addition, SD cases accounted for about 65% at the first radiological response evaluation by RECIST, which was the majority, so the focus was on SD patients, and the subsequent response and prognosis were analyzed.

As a result, in 71 patients with SD at the first radiological response evaluation by RECIST, a 25% or greater increase in the AFP value at the first radiological response evaluation from Atezo + Beva initiation was the independent affecting factor related to PD at the second radiological response evaluation on multivariate analysis. The same was also true for the 37 patients whose AFP exceeded the reference value (10 ng/mL) at the initiation of Atezo + Beva (*p* = 0.026). In the 34 patients with AFP below the reference value at the initiation of Atezo + Beva, it tended to be the independent prognostic factor (*p* = 0.059). Of them, two patients were assessed as having SD at the first radiological response evaluation and PD at the second evaluation. The AFP value in one case remained almost unchanged, but in the other case, it increased nearly twofold and exceeded the reference value at the second radiological response evaluation. All 32 cases assessed as SD at the first and second radiological response evaluations were within the reference values of AFP at the radiological second response evaluation, and none of the cases had a 25% or greater increase in AFP value. Therefore, even in cases in which the AFP value is within the reference value at treatment initiation, early detection of cases that would be assessed as PD in the future may be possible by focusing on AFP value increases.

In 59 patients with SD at the second radiological response evaluation by RECIST, a decrease in the AFP value at the second radiological response evaluation from Atezo + Beva initiation was the independent prognostic factor for PFS on multivariate analysis.

In addition, in the present study, the prognosis of SD and PD was not stratified at the first radiological response evaluation, so the cases that were SD at the second radiological response evaluation of efficacy were evaluated. If it were to be possible to analyze these cases, it would be possible to identify cases with a good prognosis at an earlier stage, and the prognosis of those cases could be improved. Furthermore, one may be able to consider early treatment modification in cases where poor response to continued treatment is anticipated in the early stages of Atezo + Beva treatment.

In this examination of 59 patients evaluated as SD at the second radiological response evaluation, there were no factors with significant differences contributing to OS. However, it has been reported that changes in ALBI score and DCP within three months after commencing Atezo + Beva affect OS [20]. Similarly, in all patients in this study, mALBI grade at the second radiological response evaluation (HR, 0.31; 95% confidence interval, 0.14–0.69; *p* = 0.0039) and a decrease in DCP values at the second radiological response evaluation from Atezo + Beva initiation (HR, 0.32; 95% confidence interval, 0.13–0.77; *p* = 0.011) were the independent prognostic factors for OS from Atezo + Beva initiation on multivariate analysis. We found no factors contributing to OS in the study of SD patients and hypothesized that the reason for the significant difference in the study of all patients is including cases with worsening liver function and poor response to Atezo + Beva within 3 months.

As a limitation of this study, AFP was not useful in mRECIST, which may be due to the lack of correlation between attenuation of hypervascularity of CT and AFP. In addition, this was a retrospective study, and the small sample size is also a limitation. Further, the observation period was relatively short, so OS could not be evaluated at the first, second, and best responses to radiological evaluations. The times of the first and second radiological response evaluations were also postponed in some cases, which is also considered a limitation.

However, it was shown that AFP is a factor associated with PD at the second radiological response evaluation by RECIST in SD patients at the first radiological response evaluation in this study. In addition, it was shown that a decrease in the AFP value at the second radiological response evaluation from Atezo + Beva initiation was related to PFS of cases judged to be SD at the first radiological response evaluation.

This suggests that, in Atezo + Beva therapy for u-HCC, AFP can be used as an index to determine whether to continue treatment.

## 5. Conclusions

The AFP trend is useful when deciding whether to continue Atezo + Beva therapy for u-HCC patients.

## Figures and Tables

**Figure 1 cancers-15-02304-f001:**
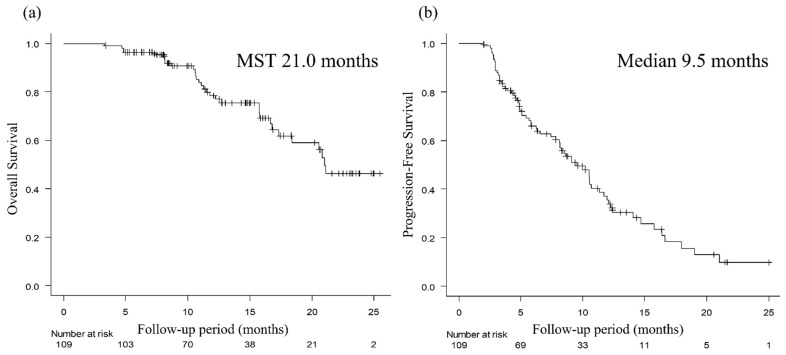
Overall survival (OS) and progression-free survival (PFS) from the initiation of atezolizumab plus bevacizumab (Atezo + Beva) in the 109 patients included in this study. (**a**) OS from the initiation of Atezo + Beva (median survival time (MST), 21.0 months). (**b**) PFS from the initiation of Atezo + Beva (median, 9.5 months).

**Figure 2 cancers-15-02304-f002:**
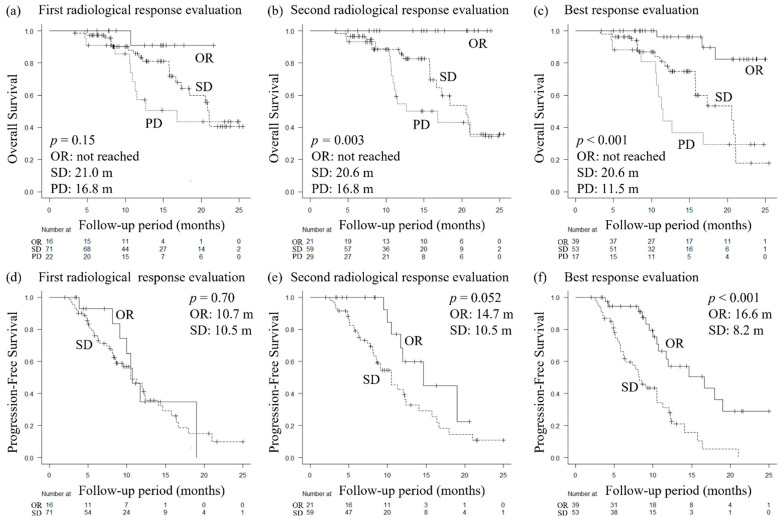
Comparison of overall survival (OS) and progression-free survival (PFS) by radiological response at the first, second, and best response evaluations by Response Evaluation Criteria in Solid Tumors (RECIST). (**a**) OS at the first response (objective response (OR) not-reached, SD 21.0 months, PD 16.8 months, *p* = 0.15). (**b**) OS at the second response (OR not-reached, SD 20.6 months, PD 16.8 months, *p* = 0.003). (**c**) OS at the best response (OR not-reached, SD 20.6 months, PD 11.5 months, *p* < 0.001). (**d**) PFS at the first response (OR 10.7 months, SD 10.5 months, *p* = 0.70). (**e**) PFS at the second response (OR 14.7 months, SD 10.5 months, *p* = 0.052). (**f**) PFS at the best response (OR 16.6 months, SD 8.2 months, *p* < 0.001).

**Figure 3 cancers-15-02304-f003:**
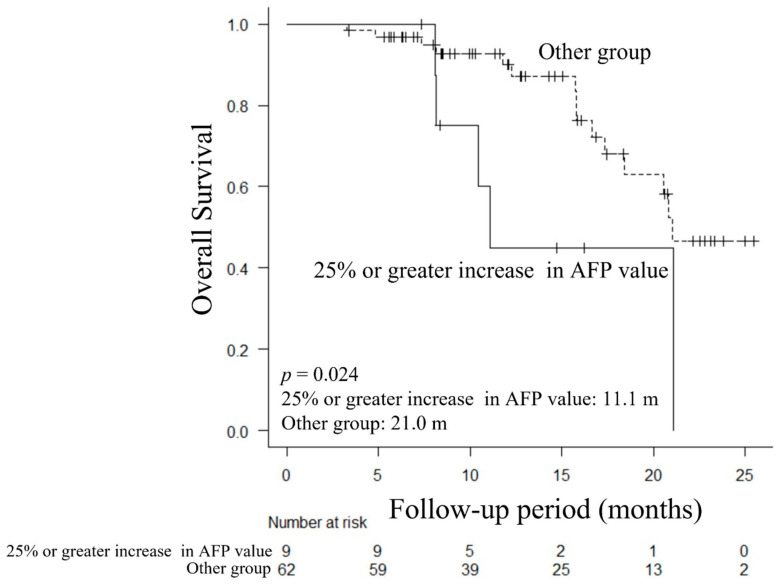
Comparison of overall survival (OS) between the group with a 25% or greater increase in AFP values at the first radiological response evaluation from Atezo + Beva initiation and the other group in 71 patients with SD at the first radiological response evaluation by Response Evaluation Criteria in Solid Tumors (25% or greater increase in AFP value 11.1 months, other group 21.0 months, *p* = 0.024).

**Table 1 cancers-15-02304-t001:** Clinical characteristic at the initiation of atezolizumab plus bevacizumab combination therapy (n = 109).

Characteristic	Median (Quartiles) or Patients, n
Age (y) *	71 (67–78)
Sex (male/female), n	90/19
Etiology (HBV/HCV/non-viral), n	17/36/56
Child-Pugh score (5/6/7), n	61/37/11
ALBI score *	−2.44 (−2.70 to −2.13)
Modified ALBI grade (1/2a/2b), n	40/32/37
Line of Atezo + Beva (1st/2nd/3rd/4th/5th), n	70/28/2/7/2
Serum ammonia level (μg/dL) *	28 (17–36)
Size of main hepatic tumor (mm) *	30 (15–50)
Relative tumor size (<50%/>50%), n	104/5
MVI (absent/present), n	88/21
EHM (absent/present), n	71/38
BCLC stage (A/B/C), n	6/52/51
Serum AFP level (ng/mL) *	24.6 (4.0–223)
Serum DCP level (mAU/mL) *	241 (77–3003)
Observation period (month) *	12.1 (8.2–16.8)

*: median (quartiles). HBV, hepatitis B virus infection; HCV, hepatitis C virus infection; ALBI, albumin–bilirubin; Atezo + Beva, atezolizumab plus bevacizumab combination therapy; MVI, macrovascular invasion; EHM, extrahepatic metastasis; BCLC, Barcelona Clinic liver cancer; AFP, alpha-fetoprotein; DCP, des-γ-carboxy prothrombin.

**Table 2 cancers-15-02304-t002:** Radiological responses to atezolizumab plus bevacizumab by Response Evaluation Criteria in Solid Tumors.

	RECIST % (n)		
	First Radiological Response Evaluation	Second Radiological Response Evaluation	Best Response Evaluation
CR	0 (0)	0 (0)	2.8 (3)
PR	14.7 (16)	19.3 (21)	33.0 (36)
SD	65.1 (71)	54.1 (59)	48.6 (53)
PD	20.2 (22)	26.6 (29)	15.6 (17)
NE	0 (0)	0 (0)	0 (0)
ORR	14.7 (16)	19.3 (21)	35.8 (39)
DCR	79.8 (87)	73.4 (80)	84.4 (92)

RECIST, Response Evaluation Criteria in Solid Tumors; CR, complete response; PR, partial response; SD, stable disease; PD, progressive disease; NE, not evaluated; ORR, overall response rate; DCR, disease control rate.

**Table 3 cancers-15-02304-t003:** Univariate and multivariate analyses of factors related to PD at the second radiological response evaluation in 71 patients with SD at the first radiological response evaluation by Response Evaluation Criteria in Solid Tumors.

Factors		Univariate		Multivariate	
		*p* Value *	Odds Ratio	95% CI	*p* Value **
Drug discontinuation before the first radiological response evaluation	absent vs. present	0.018	0.82	0.18–3.77	0.21
mALBI grade at the first radiological response evaluation	1/2a vs. 2b/3	0.060			
25% or greater in AFP value from Atezo + Beva initiation	absent vs. present	0.016	0.14	0.021–0.88	0.037
Increase in DCP value from Atezo + Beva initiation	absent vs. present	0.39			
Macroscopic vascular invasion at the first radiological response evaluation	absent vs. present	1			
Extrahepatic metastasis at the first radiological response evaluation	absent vs. present	1			
Relative tumor volume at the first radiological response evaluation	<50% vs. ≥50%	0.24			
Line of Atezo + Beva	first-line vs. later than second-line	0.070			

* Fisher’s or chi-squared test, ** Binomial logistic regression analysis. CI, confidence interval. mALBI, modified albumin–bilirubin; AFP, alpha-fetoprotein; Atezo + Beva, atezolizumab plus bevacizumab combination therapy; DCP, des-γ-carboxy prothrombin.

**Table 4 cancers-15-02304-t004:** Comparison of the rate of PD at the second radiological response evaluation in view of AFP exceeded or below the reference value (10 ng/mL) at the initiation of atezolizumab plus bevacizumab (Atezo + Beva) between the group with a 25% or greater increase in AFP values at the first radiological response evaluation from Atezo + Beva initiation and the other group.

	At the First Radiological Response Evaluation	PD % (n)	Non-PD % (n)	*p* Value *
All	25% or greater increase in AFP value	44.4 (4)	55.6 (5)	0.0017
	Other group	3.2 (2)	96.8 (60)	
AFP exceeded the reference value at the initiation of Atezo + Beva	25% or greater increase in AFP value	37.5 (3)	62.5 (5)	0.026
	Other group	3.4 (1)	96.6 (28)	
AFP below the reference value at the initiation of Atezo + Beva	25% or greater increase in AFP value	100 (1)	0 (0)	0.059
	Other group	3.0 (1)	97.0 (32)	

* Fisher’s or chi-squared test. AFP, alpha-fetoprotein; Atezo + Beva, atezolizumab plus bevacizumab combination therapy.

**Table 5 cancers-15-02304-t005:** Univariate and multivariate analyses of prognostic factors for overall survival from atezolizumab plus bevacizumab combination therapy initiation in 59 patients with SD at the second radiological response evaluation by Response Evaluation Criteria in Solid Tumors.

Factors		Univariate
		*p* Value *
Drug discontinuation before the second radiological response evaluation	absent vs. present	0.68
mALBI grade at the second radiological response evaluation	1/2a vs. 2b/3	0.085
Decrease in AFP value from Atezo + Beva initiation	absent vs. present	0.13
Decrease in DCP value from Atezo + Beva initiation	absent vs. present	0.15
Macroscopic vascular invasion at the second radiological response evaluation	absent vs. present	0.76
Extrahepatic metastasis at the second radiological response evaluation	absent vs. present	0.27
Relative tumor volume at the second radiological response evaluation	<50% vs. ≥50%	0.30
Line of Atezo + Beva	first-line vs. later than second-line	0.38

* Log-rank test. mALBI, modified albumin–bilirubin; AFP, alpha-fetoprotein; Atezo + Beva, atezolizumab plus bevacizumab combination therapy; DCP, des-γ-carboxy prothrombin.

**Table 6 cancers-15-02304-t006:** Univariate and multivariate analyses of prognostic factors for progression-free survival from atezolizumab plus bevacizumab combination therapy initiation in 59 patients with SD at the second radiological response evaluation by Response Evaluation Criteria in Solid Tumors.

Factors		Univariate		Multivariate	
		*p* Value *	Hazard Ratio	95% CI	*p* Value **
Drug discontinuation before the second radiological response evaluation	absent vs. present	0.65			
mALBI grade at the second radiological response evaluation	1/2a vs. 2b/3	0.85			
Decrease in AFP value from Atezo + Beva initiation	absent vs. present	0.040	0.46	0.23–0.89	0.022
Decrease in DCP value from Atezo + Beva initiation	absent vs. present	0.20			
Macroscopic vascular invasion at the second radiological response evaluation	absent vs. present	0.76			
Extrahepatic metastasis at the second radiological response evaluation	absent vs. present	0.53			
Relative tumor volume at the second radiological response evaluation	<50% vs. ≥50%	0.098	0.26	0.057–1.13	0.073
Line of Atezo + Beva	first-line vs. later than second-line	0.63			

* Log-rank test, ** Cox regression analysis. CI, confidence interval. mALBI, modified albumin–bilirubin; AFP, alpha-fetoprotein; Atezo + Beva, atezolizumab plus bevacizumab combination therapy; DCP, des-γ-carboxy prothrombin.

## Data Availability

The data that support the findings of this study are available from the corresponding author upon reasonable request.

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
