# Peer review of "Evaluation of Response to Atezolizumab Plus Bevacizumab in Patients with Advanced Hepatocellular Carcinoma Using the Combination of Response Evaluation Criteria in Solid Tumors and Alpha-Fetoprotein"

_cancers, 2023, doi:10.3390/cancers15082304_

Round 1

Reviewer 1 Report

Kinami et al. reported that the prognosis of patients with unresectable HCC who were treated with systemic Atezo/Bev showed that a decreased AFP level may reflect early efficacy and AFP trends could help determine whether to continue Atezo+Bev in patients assessed as having SD. The methods of analysis in this paper are fine, and I have no objection to the conclusions.

I have several comments as follows.

1. What is the reason for using a 25% increase in AFP value as one criterion in this article? Please let me know if you have any references.

2. This paper picks up several factors as predictors of PD, including 25% AFP elevation, but in Atezo+Bev therapy, treatment line has reported as a factor for anti-tumor efficacy and PFS. How do you think about the analysis of treatment line.

Reviewer 2 Report

Manuscript ID cancers-2338105 entitled " Evaluation of Response to Atezolizumab plus Bevacizumab in Patients with Advanced Hepatocellular Carcinoma using the Combination of Response Evaluation Criteria in Solid Tumors and Alpha-fetoprotein".

It should be noted that the findings of this study, which suggest that an increase of more than 25% in alpha-fetoprotein (AFP) during the second radioresponse evaluation according to Response Evaluation Criteria in Solid Tumors (RECIST) is a contributing factor linked to progressive disease (PD) in patients displaying stable disease (SD) during the initial radioresponse evaluation, as well as the reported correlation between a decrease in AFP during the second radioresponse evaluation subsequent to atezolizumab plus bevacizumab (atezo+ beva) initiation and progression-free survival (PFS) in patients categorized as having SD during the first radioresponse evaluation, may offer valuable insights for patients exhibiting early SD who are being treated with atezo+ beva. However, it is essential to recognize that this study has noteworthy limitations and concerns that necessitate further exploration.

Major comments

1.       Table 2

This research is being carried out according to the Response Evaluation Criteria in Solid Tumors (RECIST) for assessing the effects of radiation. While perusing Table 2, I noticed the mention of modified RECIST (mRECIST), however, I suggest its exclusion in order to prevent any confusion for the reader. The outcomes of mRECIST ought to be elucidated solely in the manuscript.

2.        

In HCC, AFP and DCP are critical biomarkers of the tumor. Prior research has indicated a notable correlation between elevated DCP levels and inferior OS in patients subjected to atezolizumab plus bevacizumab therapy. The absence of DCP as a prognostic determinant in the present investigation warrants comprehensive deliberation within the discussion segment.

3.        

As previously mentioned in the discussion section, the inclusion of information pertaining to the alpha-fetoprotein (AFP) levels at the initiation of atezolizumab plus bevacizumab therapy, both within and beyond the reference range (10ng/ml), would be of significant interest to the readers. Therefore, I advocate for the incorporation of these data in Figure 3.

Minor comments

1. 1. Introduction Line 67, 70, 72 and 78

Kindly unify the abbreviations to ensure consistency. Replace 'Atezo + Bev' with 'Atezo + Beva'.

2. 2. Materials and Methods 2.2. Treatments

Is it accurate to state that PD is not considered as a criterion for the discontinuation of treatment? Please check the following.

3. Table 1

It is recommended to use quartiles instead of ranges in Table 1.

4. Table 3.

Table 1 does not present any instances of mALBI 3. Therefore, the label should be amended to indicate only 2b, rather than 2b/3.

Reviewer 3 Report

In present manuscript, Kinami et al evaluated there responses of Atezolizumab plus Bevacizumab in patients with advanced HCC by using  combination of response evaluation criteria in solid tumors and alpha-fetoprotein. The author presented data showed that a decreased AFP level may reflect early efficacy, and AFP trends could help determine whether to continue Atezo + Beva in patients assessed. Overall, the author presented interesting observations might facilitate clinical practice. However, the current version must be improved as follow.

1. Language of the current form must be improved. grammatic errors and typos have seen in current version. In particular, discussion section needs reconstructed and need to be more focused. 

2. Consent regarding to patients participate in this study need to be clarified. 

3. Figures (Fig. 1. 2.3) needs replaced with high resolution images and please increase font of the label. 

4. Rational of chose AFP as response indicator need to be explained in more detail. 

5.  "atezolizumab plus bevacizumab" or "atezolizumab + bevacizumab" or "Atezo + Beva" need to be consistent. 

Round 2

Reviewer 2 Report

The primary research discoveries of this manuscript will hold significance for the findings of this investigation, which propose that an escalation exceeding 25% in alpha-fetoprotein (AFP) during the secondary radioresponse appraisal according to Response Evaluation Criteria in Solid Tumors (RECIST) constitutes a causative element connected with advancing illness (PD) in patients demonstrating stable disease (SD) during the primary radioresponse assessment. Furthermore, the reported association between a reduction in AFP during the second radioresponse evaluation following the commencement of atezolizumab plus bevacizumab (atezo+ beva) and progression-free survival (PFS) in patients classified as having SD during the initial radioresponse evaluation, might furnish valuable perspectives for patients manifesting early SD and receiving treatment with atezo+ beva.

The statistics used are appropriate and the conclusions derived from these and the figures are consistent and sound. I look forward to seeing it in print.

Reviewer 3 Report

The author have addressed concerns of this reviewer regarding to the language and style. The current version is acceptable.